# Prevalence of Rome IV Pediatric Diagnostic Questionnaire-Assessed Disorder of Gut–Brain Interaction, Psychopathological Comorbidities and Consumption of Ultra-Processed Food in Pediatric Anorexia Nervosa

**DOI:** 10.3390/nu16060817

**Published:** 2024-03-13

**Authors:** Sara Rurgo, Maria Rosaria Marchili, Giulia Spina, Marco Roversi, Flavia Cirillo, Umberto Raucci, Giovanni Sarnelli, Massimiliano Raponi, Alberto Villani

**Affiliations:** 1Department of Clinical Medicine and Surgery, University of Naples “Federico II”, 80138 Naples, Italy; sara.rurgo@unina.it (S.R.); giovanni.sarnelli@unina.it (G.S.); 2General Pediatrics and ED 2nd Level, Bambino Gesù Children’s Hospital, IRCCS, 00165 Rome, Italy; mrosaria.marchili@opbg.net (M.R.M.); marco.roversi@opbg.net (M.R.); flavia.cirillo@opbg.net (F.C.); umberto.raucci@opbg.net (U.R.); alberto.villani@opbg.net (A.V.); 3Health Directorate, Bambino Gesù Children’s Hospital, IRCCS, 00165 Rome, Italy; massimiliano.raponi@opbg.net; 4Chair of Pediatrics, Department of Systems Medicine, University of Rome “Tor Vergata”, 00133 Rome, Italy

**Keywords:** eating disorders, anorexia nervosa, functional gastrointestinal disorders, ultra-processed food, psychological problems

## Abstract

Anorexia nervosa (AN) is a severe eating disorder primarily affecting children and adolescents. Disorders of the gut–brain interaction (DGBIs) have gained recognition as significant symptoms in individuals with AN. However, limited studies have explored GI symptoms in pediatric populations with AN using age-specific diagnostic tools. This study aims to investigate the prevalence of DGBIs, their associated psychopathological aspects and their potential correlations with ultra-processed food (UPF) consumption among pediatric AN patients. The study included AN patients who were under the care of a specialized multidisciplinary team. We assessed DGBI-related symptoms using the Rome IV Pediatric Diagnostic Questionnaire on Functional Gastrointestinal Disorders (R4PDQ) and conducted psychological evaluations. Dietary intake and UPF consumption were evaluated. Among 56 AN patients, we observed a lower prevalence of DGBIs (functional constipation: 61%; functional dyspepsia: 54%; irritable bowel syndrome: 25%) compared to the existing literature. The psychological assessments revealed high rates of depression (72%) and anxiety (70%). UPF consumption was inversely related to depression levels (*p* = 0.01) but positively correlated with functional constipation (*p* = 0.046). This study highlights the importance of using age-specific diagnostic tools and emphasizes the crucial role of a specialized multidisciplinary team in the treatment of AN.

## 1. Introduction

Eating disorders (EDs), including anorexia nervosa (AN), bulimia nervosa (BN) and binge eating disorders (BED), represent a significant health concern [1,2]. Particularly affecting children, their incidence and complexity have been exacerbated by the COVID-19 pandemic [3]. Despite their increasing prevalence, the pathophysiological explanation for eating disorders remains unclear, and diagnosis is typically based on standardized international criteria [1]. 

Eating disorders are also associated with various psychiatric and somatic comorbidities and often manifest with a range of gastrointestinal symptoms, creating a complex interplay between mental health and digestive well-being [4,5,6,7]. 

There is an increasing awareness of the significance of symptoms related to gut–brain interaction disorders (DGBIs) in individuals with EDs [8]. Formerly known as functional gastrointestinal disorders (FGIDs), DGBIs represent common gastrointestinal diagnoses characterized by chronic or recurrent symptoms without structural diseases, classified using the ROME IV criteria [9]. These symptoms encompass a wide range of site-specific symptoms along the gastrointestinal (GI) tract, including abdominal pain, bloating, diarrhea and constipation, and may overlap with or exacerbate primary ED symptoms, leading to a decreased quality of life [10]. Individuals with DGBIs often experience concerns related to eating, and the connection between DGBI symptoms and EDs is increasingly acknowledged [11].

Both EDs and DGBIs hold a pivotal position within the gut–brain axis, influenced by a complex interplay of biological, psychological and social factors [12]. The scientific literature consistently reports a high prevalence of DGBIs among individuals with EDs, particularly those with AN [7]. Some studies highlight common GI symptoms in AN patients, such as postprandial fullness and abdominal pain, often influenced by psychosocial factors such as stress [13]. Notably, irritable bowel syndrome (IBS) is frequently identified as a subtype of DGBIs in AN [13]. Boyd et al. [14] found that a significant majority (98%) of ED patients met the ROME II criteria for at least one DGBI, with IBS being the most prevalent. Similarly, Santonicola et al. [5] reported a high incidence of functional dyspepsia (FD) and postprandial distress syndrome (PDS) in AN patients. Recent studies using Rome IV criteria have corroborated these findings, emphasizing the importance of assessing and managing DGBIs in ED patients, including those with restrictive food intake disorders [8,15].

Although these findings underscore the complex relationship between DGBIs and AN, there is a lack of studies examining the presence of GI symptoms in a pediatric population with ED using the age-specific Rome IV Pediatric Diagnostic Questionnaire (R4PDQ). The R4PDQ, designed for children and adolescents, offers an accurate assessment of GI symptoms relevant to this age group, considering physiological and psychological responses that may differ from adults [9].

Finally, given the complex nature of EDs, exploring the role of dietary patterns could provide valuable insights into their pathophysiology. In recent years, there has been a global increase in the consumption of ultra-processed foods (UPFs) [16,17,18,19,20,21]. 

UPFs, a category of processed foods defined by the NOVA classification, are substances found in industrial formulations like ready-to-eat meals, sugary beverages and snacks [22]. 

They are typically high in fat, sugar, added flavorings, dyes and additives, often replacing fresh, whole foods, and characterized by high levels of sugar, fat and salt, along with additives and preservatives [21]. UPFs are becoming a dominant part of dietary intake, especially among children and adolescents [23]. The consumption of UPFs has been linked to adverse health outcomes, including GI, metabolic and psychiatric issues [24]. Despite this evidence, the impact of UPFs on individuals with EDs is not yet fully understood, nor has the potential correlation between UPF consumption, DGBIs and psychopathological symptoms in EDs been explored. In keeping with this background, this study aimed to investigate (i) the prevalence of DGBIs using the specifically developed ROME IV criteria in a pediatric population with AN, (ii) the psychopathological aspects associated with the symptoms and (iii) the potential correlation with the consumption of UPFs.

## 2. Materials and Methods

### 2.1. Participants

An observational study was conducted at the Pediatric Department of Bambino Gesù Children’s Hospital, Rome, focusing on children with EDs, particularly those diagnosed with AN, referred due to severe general and nutritional status issues. This study enrolled a total of 56 patients, aged between 9 and 18 years, of both sexes who met the DSM-V criteria for AN. The selection excluded any child younger than 9 or older than 18 years, those with chronic conditions that could interfere with the study, or those who had received antibiotic treatment within the four weeks prior to the study [1]. Informed consent was obtained from all subjects involved in the study, which was approved by the local research ethics committee of Bambino Gesù Children’s Hospital under the protocol 3114_OPBG_2023, dated 16 May 2023. 

Patient management was overseen by a highly experienced multiprofessional team specializing in EDs, comprising pediatricians, child and adolescent psychiatrists, psychologists/psychotherapists, gastroenterologists, dietitians and nutrition nurses. This team facilitated a comprehensive approach to care, engaging in direct communication with both patients and their caregivers. Particular emphasis was placed on psychological support, offering patients practical coping strategies for weight recovery and managing stress during mealtimes [25]. 

### 2.2. Assessment of Gastrointestinal Symptoms

The presence of DGBI-related symptoms was evaluated using the Italian version of the Rome IV Diagnostic Questionnaire on Pediatric Functional Gastrointestinal Disorders (R4PDQ) [9]. The questionnaires systematically investigate gastrointestinal symptoms in pediatric populations. The questionnaires include several sections, each targeting different symptom clusters or specific disorders, such as functional dyspepsia, irritable bowel syndrome (IBS), functional constipation (FC) and Adolescent Rumination Syndrome, among others. The assessment utilized both parent-reported and self-report forms. Specifically, the parent-reported form was designed for children aged 4 to 10 years, recognizing that younger children might not be able to accurately articulate or self-assess their symptoms. This form allows parents or caregivers to report observations of their child’s symptoms, providing valuable insights into the child’s health that the child might not communicate. Conversely, the self-report form was employed for participants aged 11 to 18 years. This age group is generally more capable of accurately reporting their symptoms. Both forms utilized a mix of nominal and Likert scales. The nominal scales, with yes/no responses, were used to identify the presence or absence of specific gastrointestinal symptoms. The Likert scales were employed to assess symptom severity and frequency. The questionnaires’ related diagnoses are reported in the Appendix A.

### 2.3. Psychological Evaluation

Specialized psychologists administered all psychological questionnaires, conducting a daily evaluation for each patient using a specific questionnaire. In the presence of an altered score, our methodology aimed to support patients with ED, addressing each altered component, such as depression or anxiety, especially during meals (breakfast, lunch and dinner). All participants underwent the Child Depression Inventory, 2nd edition (CDI-2), a standardized Italian version consisting of 28 self-report items indicating age-specific depressive symptoms [26,27,28,29]. The Children Depression Inventory 2 (CDI 2) is a self-report questionnaire used to assess depressive symptoms in children and adolescents aged 7 to 17 years old. It consists of 28 items, each of which includes three levels of symptom severity, ranging from 0 (absent) to 2 (defined, marked). The questionnaire scores two scales of emotional and functional problems and a total score. It also provides scores on four sub-scales: negative mood/physical symptoms, negative self-esteem, ineffectiveness and interpersonal problems. Statistical analysis has shown the good quality of the test items as well as their reliability and validity in the Italian version [28]. Values are considered clinical for a T score over or equal to 70. 

Anxiety-related symptoms were assessed using the Multidimensional Anxiety Scale for Children—2nd edition self-report (MASC-2 sr), a standardized Italian questionnaire [30,31,32,33]. The MASC-2 is a self-report test designed to assess anxiety in children and adolescents aged 8 to 19 years old. The Italian version of MASC-2 has demonstrated excellent validity, good internal consistency and test–retest reliability [30]. Values are considered clinical for a T score over or equal to 70.

This 50-item questionnaire comprises six scales. Two scales are divided into two subscales, measuring the main dimensions of anxiety: anxiety from separation/phobias, the GAD Index, social anxiety (humiliation/rejection, anxiety from performance), obsessions and compulsions, physical symptoms (panic, tension/restlessness) and avoidance of danger. The MASC-2 sr produces scale scores and a total score, indicating the severity and pervasiveness of anxiety symptoms, along with an Anxiety Probability Score, assessing the possibility of having at least one anxiety disorder.

To investigate the presence of sleep disorders, participants’ parents completed the Sleep Disturbance Scale for Children (SDSC) [34,35,36]. This questionnaire evaluates behavior and sleep disorders in children aged 3 to 18 years during the previous 6 months. It consists of two sections: the first obtains demographic data, while the second comprises 26 items with a 5-point Likert scale ranging from 1 (never) to 5 (always). The score identifies a global index of disorder (SDSC TOT) and six categories of sleep disorders: disorders of onset and maintenance of sleep, respiratory sleep disorders, arousal disorders, disorders of the wake–sleep transition (DTVS), excessive daily sleepiness disorders and nocturnal hyperhidrosis.

### 2.4. Assessment of Food and Nutrient Intake

Dietary intake and consumption of UPF were measured through a 24-h recall (24HR) [37]. This approach involves a series of questions directed at the patients, covering everything they have consumed in the past 24 h, including foods, beverages, portions and condiments. Patients received guidance from a dedicated dietitian, who explained the purpose of the assessment and provided instructions on accurately remembering and recording all food and beverage intake. Patients were guided to recall and describe every item consumed during the preceding 24 h, providing detailed information such as food type, quantities, ingredients and any added condiments. To help obtain a more accurate estimate of dietary intake, we used standard portion images to quantify portion sizes. During the recording process, additional questions were asked to obtain further details about the patients’ foods and eating habits. This included questions about meal preparation, food preferences, meal timing and any snacks consumed. Once the 24HR was completed, the collected data were analyzed to determine the total caloric intake and nutritional composition of the patients’ diet [37,38]. Foods were then classified based on the NOVA classification system [22] and categorized as unprocessed, minimally processed, processed culinary ingredients, processed foods and ultra-processed foods (UPFs). 

### 2.5. Statistical Analysis

All analyses were performed with R statistical software, version 4.3.2. Continuous variables were expressed as means and standard deviations (if normally distributed) or as medians and ranges (if non-normally distributed). Categorical variables were expressed as proportions and percentages. The patients were divided into groups, according to their consumption of packaged and processed foods. Subgroup comparisons were performed with the chi-squared test for categorical variables. Two logistic regression models were built to assess the association between the significant variables from the bivariate analysis and the consumption of processed and UPF, respectively. A *p*-value less than 0.05 was considered statistically significant.

## 3. Results

A total of 56 AN patients were included in the study. The median age of the participants was 14.9 years (IQR 13.58–15.97), with a higher prevalence of females (91%). A percentage of 57.1% participated in a sport; in particular, artistic gymnastics and dance were the most reported sports. Table 1 summarizes the characteristics of the population. 

As for the psychological evaluation, the analysis of the CDI2 and MASC2 tests revealed that 72 and 70% of the patients had high scores for anxiety and depression, respectively. This finding was also associated with a poor quality of sleep, as revealed by the SDSC questionnaire, showing that 30 and 8% of the patients had mild or moderate-to-very severe sleep disorders, respectively. Table 2 summarizes the psychological evaluation of the population. 

### 3.1. Analysis of DGBI-Related Gastrointestinal Symptoms and Classification of Diagnoses According to Rome IV Criteria

Most of the patients reported GI symptoms; from the R4PDQ questionnaire, functional constipation (61%), functional dyspepsia (54%) and irritable bowel syndrome (25%) were the most prevalent diagnoses, while 9% of the patients fulfilled the criteria for rumination syndrome. As expected, the sub-analysis of patients with FD showed a high prevalence of PDS (100%), while only 10% of the subjects exhibited symptoms of epigastric pain syndrome (EPS). The GI symptoms are summarized in Table 3.

A comparison between patients accounting for a higher or lower intake of processed food is reported in Table 4.

We found that only a few variables were significantly different between the patients with a higher (≥2 meals/day) or lower (0–1 meals/day) intake of processed food, namely the mean value of the domain “P” of the MASC-2 score (10.87 vs. 7.59, *p* = 0.035), the consumption for breakfast (96.7% vs. 69.2%, *p* = 0.039) and the consumption for dinner (27.6% vs. 0%, *p* = 0.048).

A comparison between patients accounting for a higher or lower intake of UPFs is reported in Table 5.

When we compared patients with a higher (≥2 meals/day) or lower (0–1 meals/day) intake of UPFs, we found that the former would practice sport less than the latter (33.3% vs. 68.4%, *p* = 0.029). We also found a significantly different distribution of the risk of depression according to the CDI-2 score, with a higher risk among patients who consumed more UPFs (*p* = 0.011). These kinds of patients were also more prone to sleep disorders, with significantly higher scores at the SDSC (see Table 4). The consumption of UPFs was also more frequent for breakfast (77.8% vs. 24.0%, *p* = 0.001), morning snacks (85.7% vs. 27.3%, *p* = 0.011) and afternoon snacks (94.1% vs. 29.4%, *p* < 0.001).

The logistic regression model adopting the consumption of processed and UPFs in ≥2 meals/day as the dependent variable is shown in Table 6 and Table 7, respectively.

Interestingly, practicing no sport was the only variable significantly associated with a higher consumption of ultra-processed food.

### 3.2. Consumption of UPFs and Correlation with Psychological Features and DGBI

An overall analysis revealed that the majority of the patients consumed processed foods and UPFs at least once a day. In particular, 76.5% and 61.8% of them referred to consuming PF and UPFs as snacks throughout the entire day, respectively. 

In our population, we found a significantly higher intake of UPFs in subjects with more severe depression scores (*p* = 0.01), while no significant difference emerged as far as the anxiety component, nor was sleep quality associated with the consumption of UPFs. 

To further investigate the relationship between UPF consumption and GI symptoms, we calculated the Cohen’s K of the confusion matrices for each GI symptom. From this analysis, we found that only FC was mildly concordant with a higher level of assumption of UPFs (Cohen’s K = 0.214, *p* = 0.046). No significant differences emerged by considering other DGBIs.

## 4. Discussion

This study analyzed the prevalence of DGBIs and psychological conditions in a pediatric population with AN. The research also focused on the patients’ dietary habits, with a specific look at the consumption of UPFs. 

Recognizing and managing DGBI symptoms in patients with AN is a crucial aspect of clinical practice; untreated DGBIs can distract from the primary pathology of an ED, perpetuate disordered eating behaviors and hinder nutritional rehabilitation. A comprehensive evaluation to rule out structural or organic causes is essential, and management may involve reassurance, neuromodulators and complementary therapies [15,39].

In our study population, functional constipation (FC) and FD with PDS were the most often described. Irritable bowel syndrome (IBS) was reported at a percentage rate of 25.5%. It is crucial to emphasize that we observed a lower prevalence of DGBIs compared to the existing literature. This is likely attributed to the use of the R4PDQ, a more sensitive and specific tool. 

The existing literature consistently reports a higher prevalence of DGBIs in patients with AN. For instance, in a study of 85 adult outpatients with AN, according to the ROME III criteria, 90% met the criteria for PDS, and 93% reported constipation-type IBS-C, with a higher prevalence observed in those with a lower body mass index (BMI) and longer disease durations of over 5 years [5]. Boyd et al. [14] found that a significant percentage (98%) of individuals with EDs, including 44% with AN, met the ROME II criteria for at least one DGBI, with IBS being most common. Finally, a recent study specifically investigated the prevalence of DGBIs in patients with AN according to the Rome IV criteria. The research found that 97.4% of the sample met the diagnostic criteria for FD, of which 88.8% presented the PDS subtype and 41.6% presented the EPS subtype. In addition, 52.6% of the sample met the diagnostic criteria for IBS, while for FC, the prevalence reached 7.9% [8]. 

This discrepancy in the observed prevalence may be attributed to our use of the R4PDQ, which is an age-specific and standardized tool for assessing GI symptoms in pediatric populations. These mentioned studies focused, in fact, on the broader Rome criteria, which are a set of standardized guidelines used to diagnose functional gastrointestinal disorders but are not necessarily confined to these specific pediatric diagnostic questionnaires. The R4PDQ is specifically designed for children and adolescents. This age-specific focus might result in different prevalence rates compared to studies that use the broader Rome criteria, which are applied across a wider age range. The pediatric questionnaires may have different sensitivities and specificities for identifying DGBIs in children and adolescents compared to the general Rome IV criteria. This could lead to variations in the reported prevalence of DGBIs among different studies.

One study, conducted among schoolchildren aged 10 to 18 years in Colombia, found that the R4PDQ had a sensitivity of 75% and a specificity of 90%, suggesting that the R4PDQ has adequate diagnostic accuracy for diagnosing DGBIs in children [40]. Recently, Strisciuglio C. et al. [41] used the R4PDQ to assess the prevalence of DGBIs among children and adolescents in Mediterranean countries. This was a key methodological aspect of their research, allowing for a standardized assessment of DGBIs according to the latest diagnostic criteria. The results indicate variations in the prevalence of DGBIs among different European countries and a comparison between the Rome IV and Rome III criteria, suggesting a lower prevalence of DGBIs in children using R4PDQ. Moreover, Kaul I. et al. [42] suggest that there is a higher level of agreement among pediatric gastroenterologists in diagnosing DGBIs in children using the Rome IV criteria than in choosing diagnostic tests. While there is a 68% agreement rate in diagnoses based on the Rome IV criteria, the agreement on diagnostic testing is less than 30%. This indicates that despite the widespread adoption of the Rome IV criteria in clinical practice, there is still significant variability in the selection of specific diagnostic tests for DGBIs in children. 

Finally, our hypothesis is that the lower prevalence of DGBIs observed in our population suggests that a multidisciplinary approach may play a role in managing patients’ symptoms. It is important to underline that GI symptoms in patients with AN are often linked to malnutrition or purging behaviors. Addressing these factors can frequently lead to an improvement in patients’ GI symptoms. In our own clinical practice, patients receive comprehensive care aimed at restoring proper nutritional intake, motivating patients towards spontaneous eating, and reducing concurrent psychiatric symptoms [25]. The correction of eating behaviors through clinical, nutritional and psychological interventions and the reduction in psychiatric comorbidities through pharmacological therapy may help the management of GI symptoms and, consequently, could explain the decreased prevalence of GI symptoms seen in AN patients during their hospitalization in our department [43]. However, studies are needed to support our hypothesis.

Our findings contribute valuable insights into the relationship between DGBIs and AN in pediatric populations and pave the way for further research using age-appropriate diagnostic tools like the R4PDQ to better understand this relationship in younger populations. This could lead to more tailored approaches to the diagnosis and management of DGBIs in children and adolescents with AN. 

As for the psychological impairment described in our population and the high rates of depression and anxiety, we strongly recommend our kind of methodology, characterized by meal assistance and supervision. Psychological care focuses on providing patients with functional coping strategies for weight recovery and stress management during mealtimes. Recent research is in line with our hypothesis, demonstrating a positive impact on ED considering eating behavior and dysfunctional attitudes [43]. Our study also focused on sleep disturbances, describing a normal rate of sleep in the majority of the population (60.9%). This could probably be connected to the parents’ questionnaire compilation, which sometimes does not reflect patients’ perceptions. In addition, the literature has reviewed sleep problems and food quality intake, concluding that there is an urge for more research to investigate the influence of sleep on eating habits [44]. 

It was found that UPF consumption was low among these patients, many of whom experienced starvation. The use of the 24HR in the context of hospitalization likely led to an underestimation of the patients’ usual intakes, as they were in an acute condition that in some cases also required the use of enteral nutrition through a nasogastric tube. This represents a limitation of our study. A noteworthy aspect of our findings is the inverse relationship between depression levels and UPF consumption in AN patients. Patients with elevated depression scores (high CDI2 scores) tend to consume fewer UPFs. This can be explained by the prevalence of emotional and behavioral symptoms typical of acute AN and depression, which are characterized by food avoidance due to the drive for thinness and negative self-image [43]. These aspects drive patients with AN to give up almost all types of foods, including UPFs.

The high rate of depression among these patients correlates with a reduced interest in food and, consequently, with reduced UPF intake. As for DGIBIs, a direct correlation emerged between the consumption of UPFs and FC; in particular, the patients with FC declared that they consume more UPFs. It is likely that the low fiber content of UPFs could be involved in the induction and/or exacerbation of constipation [45]. No correlation was found between other GI symptoms and UPF consumption. This trend aligns with the severity of EDs observed in patients admitted for severe malnutrition, necessitating acute interventions like nasogastric feeding. The reluctance or avoidance to consume UPFs could be indicative of the severity of the anorexic condition, where the fear of weight gain outweighs the convenience and appeal of these foods [45].

Moreover, this could be explained by the hypothesis that GI symptoms in patients with acute AN are often somatic expressions of a psychogenic nature. This is related to a diminished ability to recognize and regulate one’s emotions, as well as a reduced capacity for introspection [46,47]. Furthermore, it is conceivable that gastrointestinal symptoms are frequently used as strategies to avoid food intake. Our results are in line with the existing literature; a recent study aimed to investigate the relationship between UPF consumption and disordered eating patterns. It focused on patients with AN, BN and BED, exploring the prevalence of UPFs in their diet using the NOVA classification system. The study found that the patients with AN reported a lower consumption of UPFs compared to those with BN and BED [48]. 

This study presents both strengths and weaknesses in its design and methods. As a strong point, age-specific diagnostic questionnaires like R4PDQ were used in order to assess DGBIs in pediatric AN patients. Another strength is found in the extensive caseload of a tertiary care hospital specializing in disorders like AN and offering comprehensive multidisciplinary management guaranteed by the expertise and experience of a multispecialist team. However, the study is limited in its ability to establish cause–effect relationships due to its observational nature, focusing only on associations. Moreover, the use of self-reported data raises concerns about the subjective nature of the information, which could affect the study’s overall reliability and applicability. Particularly, the accuracy of 24HR as a measure introduces limitations as it relies on the participants’ memory and honesty, which can lead to inaccuracies in reporting dietary intake. The single site (a tertiary pediatric hospital) of our study may also limit the generalizability of the results. For this reason, our results and conclusions should be wisely compared and generalized to patients treated in different settings.

These methodological limitations underline the need for more rigorous designs in future research to overcome such challenges.

The future goal is to compare the effect of psychological support on eating behaviors from admission to discharge while implementing psychological treatment and probably also improving DGBIs. 

## 5. Conclusions

In contrast to the existing literature, this study highlights a lower prevalence of DGBIs in pediatric patients with AN when using the R4PDQ. Our results support the importance of age-specific diagnostic tools and the importance of a specialized multidisciplinary team, particularly in mealtime assistance and stress management. These insights enhance our understanding of AN in pediatric populations. In addition, the study suggests the need for further research into the impact of UPFs on DGBIs and psychological symptoms. While we recommend our methodology for potential improvements in DGBIs, further research is needed for validation.

## Figures and Tables

**Table 1 nutrients-16-00817-t001:** Characteristics of the population.

Characteristics	Values—No. (%)
Females	51 (91)
Sport practice	32 (57)
Type of sport practices	
Gymnastics/athletics	7 (22)
Dance	6 (19)
Swimming	5 (16)
Karate	3 (9)
Gym	3 (9)
Soccer	2 (6)
Volleyball	2 (6)
Tennis	1 (3)
Other	4 (13)

**Table 2 nutrients-16-00817-t002:** Psychological evaluation according to the CDI-2, MASC-2 and SDSC tests.

Psychological Evaluation	Values
CDI-2 (domains)	
Physical symptoms of negative mood—median [IQR]	8.00 [5.00, 10.00]
Self-esteem—median [IQR]	7.00 [4.00, 8.00]
Inefficacy—median [IQR]	7.00 [5.00, 9.50]
Interpersonal problems—means (SD)	3.89 (2.28)
Emotional problems—mean (SD)	13.02 (6.09)
Functional problems—mean (SD)	10.70 (5.29)
CDI-2 score (raw)—mean (SD)	23.72 (11.01)
CDI-2 score (age-normalized)—mean (SD)	65.15 (13.16)
CDI-2 score (classification)	
Low/moderate risk	13 (28)
Over average risk	7 (15)
High risk	8 (17)
Very high risk	19 (40)
MASC-2 self-reported (domains)	
SP—median [IQR]	8.00 [5.00, 14.00]
GAD—median [IQR]	17.00 [13.00, 21.50]
SA.T—median [IQR]	19.00 [11.00, 23.50]
HR—median [IQR]	10.00 [6.00, 13.50]
PF–median [IQR]	9.00 [5.50, 10.00]
OC—mean (SD)	13.21 (8.27)
PS.T—mean (SD)	18.96 (8.34)
P—mean (SD)	9.68 (5.16)
TR—median [IQR]	10.00 [7.00, 12.50]
HA—median [IQR]	17.00 [15.00, 19.50]
MASC-2 score (raw)—mean (SD)	78.04 (27.54)
MASC-2 score (age-normalized)—mean (SD)	64.98 (15.46)
MASC-2 score (classification)	
Average risk	14 (30)
Average/high risk	1 (2)
Moderately high risk	8 (17)
High risk	7 (15)
Very high risk	17 (36)
SDSC (domains)	
DIMS—median [IQR]	14.00 [12.00, 18.75]
DRS—median [IQR]	3.00 [3.00, 4.00]
DA—median [IQR]	3.00 [3.00, 4.75]
DTVS—median [IQR]	8.00 [6.25, 10.00]
DES—median [IQR]	9.00 [5.00, 13.75]
IPN—mean (SD)	2.48 (1.09)
SDSC score (raw)—mean (SD)	45.48 (14.90)
SDSC score (classification)—no. (%)	
Normal	28 (61)
Mild	14 (30)
Moderate	2 (4)
Severe	1 (2)
Very severe	1 (2)

CDI-2, Child Depression Inventory—2nd edition; MASC-2, Multidimensional Anxiety Scale for Children—2nd edition self-report; SDSC, Sleep Disturbance Scale for Children; SPs, separation phobias; GAD, Generalized Anxiety Disorder; SA.T, social anxiety; HR, humiliation/rejection; PF, performance anxiety; OCs, obsessions and compulsions; PS.T, physical symptoms; P, panic; TR, tension/restlessness; HA, avoidance of danger; DIMS, sleep initiation and maintenance disorder; DRSs, Sleep-Related Breathing Disorders; DAs, arousal disorders; DTVSs, sleep–wake transition disorders; DES, excessive sleepiness disorder; IPN, nocturnal hyperhidrosis.

**Table 3 nutrients-16-00817-t003:** Gastrointestinal symptoms according to Rome IV Criteria.

R4PDQ-DGBIs	No. (%)
FC	34 (61)
FD	30 (54)
PDS	30 (100)
EPS	3 (10)
IBS	14 (25)
Rumination syndrome	5 (9)
Aerophagia	3 (5)
Functional vomiting	1 (2)

DGBI, disorders of gut–brain interaction; R4PDQ, Rome IV Diagnostic Questionnaire on Pediatric Functional Gastrointestinal Disorders; FC, functional constipation; FD, functional dyspepsia; PDS, postprandial distress syndrome; EPS, epigastric pain syndrome; IBS, irritable bowel syndrome.

**Table 4 nutrients-16-00817-t004:** Comparison according to processed food intake.

	≥2 Meals/Day	0–1 Meals/Day	*p*-Value
Age (years)—median [IQR]	15.04 [14.01, 16.01]	14.87 [12.51, 15.80]	0.240
Females—no. (%)	27 (90.0)	24 (92.3)	1.000
Sport practice—no. (%)	14 (46.7)	18 (69.2)	0.152
Sport practice (times per week)—median [IQR]	3.00 [2.00, 3.75]	2.00 [2.00, 3.75]	0.315
DGBI-related gastrointestinal symptoms—no. (%)			
FC	22 (73.3)	12 (57.1)	0.365
FD	15 (50.0)	14 (58.3)	0.737
PDS	15 (50.0)	15 (62.5)	0.520
EPS	2 (8.3)	1 (7.7)	1.000
IBS	10 (33.3)	4 (16.0)	0.247
Rumination syndrome	4 (40.0)	1 (14.3)	0.546
Aerophagia	2 (6.7)	1 (5.0)	1.000
Functional vomiting	1 (14.3)	0 (0)	1.000
CDI-2 (domains)	9.00 [6.00, 10.00]	5.00 [3.00, 9.00]	0.155
Physical symptoms of negative mood—median [IQR]	7.00 [4.00, 8.00]	5.00 [3.00, 7.00]	0.222
Self-esteem—median [IQR]	8.00 [6.00, 10.00]	7.00 [2.00, 8.00]	0.189
Inefficacy—median [IQR]	4.33 (2.23)	3.12 (2.20)	0.078
Interpersonal problems—means (SD)	13.97 (5.80)	11.35 (6.40)	0.160
Emotional problems—mean (SD)	11.63 (5.22)	9.06 (5.14)	0.109
Functional problems—mean (SD)	25.60 (10.65)	20.41 (11.16)	0.122
CDI-2 score (raw)—mean (SD)	66.10 (13.27)	63.47 (13.20)	0.517
CDI-2 score (age-normalized)—mean (SD)	9.00 [6.00, 10.00]	5.00 [3.00, 9.00]	0.155
CDI-2 score (classification)			
Low/moderate risk	8 (26.7)	5 (29.4)	0.346
Over average risk	3 (10.0)	4 (23.5)	
High risk	7 (23.3)	1 (5.9)	
Very high risk	12 (40.0)	7 (41.2)	
MASC-2 self-reported (domains)			
SP—median [IQR]	9.00 [5.00, 14.75]	8.00 [5.00, 13.00]	0.363
GAD—median [IQR]	18.50 [15.25, 22.00]	16.00 [11.00, 18.00]	0.137
SA.T—median [IQR]	21.00 [13.50, 23.75]	16.00 [11.00, 19.00]	0.340
HR—median [IQR]	11.00 [7.50, 14.00]	8.00 [5.00, 10.00]	0.114
PF—median [IQR]	9.00 [6.00, 10.00]	9.00 [5.00, 10.00]	0.789
OC—mean (SD)	13.30 (8.05)	13.06 (8.89)	0.925
PS.T—mean (SD)	20.67 (7.47)	15.94 (9.15)	0.061
P—mean (SD)	10.87 (4.48)	7.59 (5.73)	0.035
TR—median [IQR]	11.00 [7.00, 13.00]	8.00 [4.00, 12.00]	0.190
HA—median [IQR]	17.00 [14.25, 18.75]	18.00 [15.00, 20.00]	0.430
MASC-2 score (raw)—mean (SD)	80.90 (26.99)	73.00 (28.61)	0.350
MASC-2 score (age-normalized)—mean (SD)	66.93 (15.34)	61.53 (15.52)	0.254
MASC-2 score (classification)			
Average risk	7 (23.3)	7 (41.2)	0.323
Average/high risk	0 (0)	1 (5.9)	
Moderately high risk	6 (20.0)	2 (11.8)	
High risk	4 (13.3)	3 (17.6)	
Very high risk	13 (43.3)	4 (23.5)	
SDSC (domains)			
DIMS—median [IQR]	16.00 [11.00, 22.00]	13.00 [12.00, 16.00]	0.855
DRS—median [IQR]	3.00 [3.00, 4.00]	3.00 [3.00, 4.00]	0.142
DA—median [IQR]	3.00 [3.00, 5.00]	3.00 [3.00, 4.00]	0.225
DTVS—median [IQR]	8.00 [6.00, 10.00]	9.00 [7.00, 10.00]	0.620
DES—median [IQR]	9.00 [7.00, 14.00]	6.00 [5.00, 13.00]	0.189
IPN—mean (SD)	2.55 (1.21)	2.35 (0.86)	0.556
SDSC score (raw)—mean (SD)	46.66 (16.56)	43.47 (11.72)	0.490
SDSC score (classification)—no. (%)			
Normal	16 (55.2)	12 (70.6)	0.168
Mild	11 (37.9)	3 (17.6)	
Moderate	0 (0)	2 (11.8)	
Severe	1 (3.4)	0 (0)	
Very severe	1 (3.4)	0 (0)	
For breakfast—no. (%)	29 (96.7)	9 (69.2)	0.039
For morning snack—no. (%)	17 (81.0)	1 (25.0)	0.094
For lunch—no. (%)	8 (26.7)	0 (0)	0.053
For afternoon snack—no. (%)	23 (88.5)	3 (37.5)	0.013
For dinner—no. (%)	8 (27.6)	0 (0)	0.048
For other—no. (%)	3 (37.5)	1 (33.3)	1.000

DGBI, disorders of gut–brain interaction; FC, functional constipation; FD, functional dyspepsia; PDS, postprandial distress syndrome; EPS, epigastric pain syndrome; IBS, irritable bowel syndrome; CDI-2, Child Depression Inventory—2nd edition; MASC-2, Multidimensional Anxiety Scale for Children—2nd edition self-report; SDSC, Sleep Disturbance Scale for Children; SP, separation phobias; GAD, Generalized Anxiety Disorder; SA.T, social anxiety; HR, humiliation/rejection; PF, performance anxiety; OCs, obsessions and compulsions; PS.T, physical symptoms; P, panic; TR, tension/restlessness; HA, avoidance of danger; DIMS, sleep initiation and maintenance disorder; DRSs, Sleep-Related Breathing Disorders; DAs, arousal disorders; DTVSs, sleep–wake transition disorders; DES, excessive sleepiness disorder; IPN, nocturnal hyperhidrosis.

**Table 5 nutrients-16-00817-t005:** Comparison according to UPF intake.

	≥2 Meals/Day	0–1 Meals/Day	*p*-Value
Age (years)—median [IQR]	14.85 [14.01, 16.01]	15.09 [13.14, 15.80]	0.511
Females—no. (%)	15 (83.3)	36 (94.7)	0.370
Sport practice—no. (%)	6 (33.3)	26 (68.4)	0.029
Sport practice (times per week)—median [IQR]	3.00 [2.25, 3.75]	2.00 [2.00, 3.75]	0.450
DGBI-related gastrointestinal symptoms—no. (%)			
FC	15 (83.3)	19 (57.6)	0.120
FD	10 (55.6)	19 (52.8)	1.000
PDS	10 (55.6)	20 (55.6)	1.000
EPS	2 (13.3)	1 (4.5)	0.728
IBS	6 (33.3)	8 (21.6)	0.545
Rumination syndrome	1 (20.0)	4 (33.3)	1.000
Aerophagia	1 (5.6)	2 (6.2)	1.000
Functional vomiting	1 (20.0)	0 (0)	1.000
CDI-2 (domains)			
Physical symptoms of negative mood—median [IQR]	9.00 [6.25, 10.00]	8.00 [3.00, 10.00]	0.191
Self-esteem—median [IQR]	7.00 [5.00, 8.00]	5.00 [3.00, 7.00]	0.195
Inefficacy—median [IQR]	7.50 [6.00, 10.00]	7.00 [2.00, 8.00]	0.266
Interpersonal problems—means (SD)	4.33 (1.71)	3.62 (2.56)	0.302
Emotional problems—mean (SD)	14.72 (4.30)	11.97 (6.83)	0.133
Functional problems—mean (SD)	12.00 (3.74)	9.90 (5.97)	0.188
CDI-2 score (raw)—mean (SD)	26.72 (7.40)	21.86 (12.51)	0.143
CDI-2 score (age-normalized)—mean (SD)	68.00 (10.63)	63.38 (14.41)	0.246
CDI-2 score (classification)			
Low/moderate risk	3 (16.7)	10 (34.5)	0.011
Over average risk	1 (5.6)	6 (20.7)	
High risk	7 (38.9)	1 (3.4)	
Very high risk	7 (38.9)	12 (41.4)	
MASC-2 self-reported (domains)			
SP—median [IQR]	8.00 [3.25, 14.00]	9.00 [5.00, 14.00]	0.599
GAD—median [IQR]	18.00 [16.00, 22.00]	17.00 [11.00, 21.00]	0.293
SA.T—median [IQR]	20.50 [13.50, 22.75]	18.00 [11.00, 24.00]	0.843
HR—median [IQR]	10.00 [7.50, 13.75]	10.00 [5.00, 13.00]	0.652
PF—median [IQR]	8.50 [6.25, 10.00]	9.00 [5.00, 10.00]	0.965
OC—mean (SD)	12.72 (6.82)	13.52 (9.16)	0.753
PS.T—mean (SD)	21.44 (7.06)	17.41 (8.81)	0.108
P—mean (SD)	11.33 (5.05)	8.66 (5.04)	0.084
TR—median [IQR]	10.50 [7.50, 12.00]	9.00 [4.00, 13.00]	0.442
HA—median [IQR]	15.50 [12.25, 18.75]	18.00 [16.00, 20.00]	0.141
MASC-2 score (raw)—mean (SD)	79.39 (23.79)	77.21 (30.01)	0.795
MASC-2 score (age-normalized)—mean (SD)	66.56 (13.69)	64.00 (16.62)	0.587
MASC-2 score (classification)			
Average risk	4 (22.2)	10 (34.5)	0.246
Average/high risk	0 (0)	1 (3.4)	
Moderately high risk	5 (27.8)	3 (10.3)	
High risk	1 (5.6)	6 (20.7)	
Very high risk	8 (44.4)	9 (31.0)	
SDSC (domains)			
DIMS—median [IQR]	17.00 [12.00, 22.00]	13.00 [12.00, 18.00]	0.386
DRS—median [IQR]	3.00 [3.00, 4.00]	3.00 [3.00, 4.00]	0.968
DA—median [IQR]	4.00 [3.00, 6.00]	3.00 [3.00, 4.00]	0.015
DTVS—median [IQR]	10.00 [7.00, 11.00]	8.00 [6.00, 10.00]	0.200
DES—median [IQR]	10.00 [8.00, 17.00]	7.00 [5.00, 13.00]	0.042
IPN—mean (SD)	2.94 (1.48)	2.21 (0.68)	0.026
SDSC score (raw)—mean (SD)	50.76 (17.99)	42.38 (12.03)	0.065
SDSC score (classification)—no. (%)			
Normal	8 (47.1)	20 (69.0)	0.168
Mild	7 (41.2)	7 (24.1)	
Moderate	0 (0)	2 (6.9)	
Severe	1 (5.9)	0 (0)	
Very severe	1 (5.9)	0 (0)	
For breakfast—no. (%)	14 (77.8)	6 (24.0)	0.001
For morning snack—no. (%)	12 (85.7)	3 (27.3)	0.011
For lunch—no. (%)	1 (5.6)	1 (3.4)	1.000
For afternoon snack—no. (%)	16 (94.1)	5 (29.4)	<0.001
For dinner—no. (%)	2 (11.8)	0 (0)	0.254
For other—no. (%)	3 (50.0)	1 (20.0)	0.689

DGBI, disorders of gut–brain interaction; FC, functional constipation; FD, functional dyspepsia; PDS, postprandial distress syndrome; EPS, epigastric pain syndrome; IBS, irritable bowel syndrome; CDI-2, Child Depression Inventory—2nd edition; MASC-2, Multidimensional Anxiety Scale for Children—2nd edition self-report; SDSC, Sleep Disturbance Scale for Children; SP, separation phobias; GAD, Generalized Anxiety Disorder; SA.T, social anxiety; HR, humiliation/rejection; PF, performance anxiety; OCs, obsessions and compulsions; PS.T, physical symptoms; P, panic; TR, tension/restlessness; HA, avoidance of danger; DIMS, sleep initiation and maintenance disorder; DRSs, Sleep-Related Breathing Disorders; DAs, arousal disorders; DTVSs, sleep–wake transition disorders; DES, excessive sleepiness disorder; IPN, nocturnal hyperhidrosis.

**Table 6 nutrients-16-00817-t006:** Logistic regression model (consumption of processed food ≥2 meals/day).

	OR	95% C.I.	*p*-Value
Age (years)	0.770	0.538–1.102	0.153
Sex (male)	0.413	0.037–4.582	0.471
MASC-2 score (raw)	0.989	0.966–1.013	0.362

MASC-2, Multidimensional Anxiety Scale for Children—2nd edition self-report.

**Table 7 nutrients-16-00817-t007:** Logistic regression model (consumption of ultra-processed food ≥2 meals/day).

	OR	95% C.I.	*p*-Value
Age (years)	0.664	0.378–1.165	0.153
Sex (male)	0.033	0.001–1.295	0.068
Sport (no)	0.148	0.033–0.674	0.013
CDI-2 score (raw)	0.990	0.906–1.082	0.822
SDSC score (raw)	0.949	0.882–1.022	0.168

CDI-2, Child Depression Inventory—2nd edition; SDSC, Sleep Disturbance Scale for Children.

## Data Availability

Requests for access to data, statistical codes, questionnaires and technical processes may be made by contacting the corresponding author at giulia.spina@opbg.net.

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
