# Peer review of "Prevalence of Rome IV Pediatric Diagnostic Questionnaire-Assessed Disorder of Gut–Brain Interaction, Psychopathological Comorbidities and Consumption of Ultra-Processed Food in Pediatric Anorexia Nervosa"

_nutrients, 2024, doi:10.3390/nu16060817_

Round 1

Reviewer 1 Report

Comments and Suggestions for Authors

Excellent research. Very well-structured and developed, so I only have some minor recommendations to make:

  1. 1. In section 2.1. Participants, it does not specify the sample size nor define the study population correctly, so both issues should be addressed in that section. Some of the missing information is at the beginning of the Results section but should be moved to this section.
  2. 2. In section 2.2. Assessment of Gastrointestinal Symptoms, the psychometric properties of reliability and validity of the instrument should be specified, along with describing its configuration and most notable characteristics.
  3. 3. In section 2.3. Psychological Evaluation, the reliability of the instrument needs to be determined (for example, using Cronbach's alpha).
  4. 4. In the Results section, I recommend including at least one new table with inferential statistical analyses since only descriptive tables are included, and the inferential analyses (which are very important) are presented only in text form in a small paragraph.

Reviewer 2 Report

Comments and Suggestions for Authors

COMMENTS TO THE AUTHORS

 Reference: Manuscript ID: nutrients-2875949

 It is highlighted in blue and italics, the parts of the text that are exposed in a literal way.

The paper titledTitle: Prevalence of Rome IV Pediatric Diagnostic Question-naires-assessed Disorder of Gut-Brain Interaction, Psycho-pathological Comorbidities and consumption of Ul-tra-processed food in Pediatric anorexia Nervosa, addresses an interesting topic.

On page 2, lines 79-90, the authors show the objectives of the research as follows:

They are typically high in fat, sugar, added flavorings, dyes, and additives, often replacing fresh, whole foods and characterized by high levels of sugar, fat, and salt, along  with additives and preservatives [21]. UPFs are becoming a dominant part of dietary intake especially in children and adolescents [23]. The consumption of UPFs has been  linked to adverse health outcomes, including GI, metabolic and psychiatric issues [24].Despite this evidence, the impact of UPFs on individuals with ED is not yet fully under stood, nor the potential correlation between UPFs consumption, DGBIs, and psychopathological symptoms in ED has been explored. In keeping with this background, this study aimed to investigate i) the prevalence of DGBIs, using the specifically developed-ROME IV criteria, in a pediatric population with AN, ii) the psychopathological aspects associated that with the symptoms, and iii) to explore the potential correlation with the consumption of ultraprocessed food.

The topic they address is very interesting. However, the research has been carried out without any experimental rigor, the lack of control is absolute, the data analysis needs to be more robust, and it has not been possible to carry out statistical control of confusion variables. In more detail: 

A.- On page 3, lines 79-90, 2.1. Los autores describen:

2.1.- Participants An observational study was conducted, involving children with ED referred to the Pediatric Department at Bambino Gesù Children's Hospital, Rome, due to severe general and nutritional status. Consecutive patients diagnosed with AN according to DSM-V criteria and with the following inclusion criteria were enrolled: age 9-18 years, both sexes, absence of alarm symptoms suggestive of an organic GI disease were enrolled [1]. Exclusion criteria were: age < 9 and > 18 years, concurrent presence of chronic pathologies, and 98 antibiotic treatment in the previous four weeks. 

The age variable is crucial in the topic discussed by the authors. Between 9 and 18 are the three stages of pre-adolescence and adolescence, and it is probably the most turbulent age of a person's existence. This variable should be a control variable in this research. It must be controlled statistically (covariance analysis) and/or through subgroup analysis by age range (three age ranges could be made, for example). This is not done at any time. Therefore, the variability in response due to the influence of age dissolves. Thus, the data analysis must be completely redone to consider this aspect. I will insist on this aspect again later

B.- The authors must describe the evaluation instruments they used in more detail. In section 2.2. Assessment of Gastrointestinal Symptoms

Presence of DGBI-related symptoms was evaluated using the Italian version of the Rome IV Diagnostic Questionnaire on Pediatric Functional Gastrointestinal Disorders (R4PDQ) [9]. Questionnaires systematically investigated gastrointestinal symptoms and disorders in pediatric populations. The assessment utilized both self-report and parent reported forms. The questionnaires’ related diagnoses are reported in the supplementary material. 

-What was the purpose of both self-report and parent reported forms? Please explain and provide information on what the measurement was like (Likert scale? 

In section 2.3. Psychological Evaluation: 

- For example, in the Child Depression Inventory, 2nd edition (CDI-2), and the Multidimensional Anxiety Scale for Children–2nd edition self-report (MASC-2 sr), is the response recorded on a Likert scale? How much is its maximum score? Is there a measure that indicates when the response indicates a severe pathology? 

-To investigate the presence of sleep disorders, participants' parents completed the Sleep Disturbance Scale for Children (SDSC) [34-36]. Are the parents the ones who answer? Is it reliable that the parents answer this questionnaire?

-Similarly, Anxiety-related symptoms were assessed using the Multidimensional Anxiety Scale for Children–2nd edition self-report (MASC-2 sr), a standardized Italian questionnaire [30-33]. This 50-item questionnaire, filled out by the person evaluating filled out by the person evaluating, the ones who fill out this scale? Is the answer reliable? 

The two previous points must have an explanation.

In section 2.4. Assessment of Food and Nutrient Intake: 

Dietary intake and consumption of UPFs was measured through a 24-hour recall (24HR) completed with trained research assistants, as previously reported [37, 38]. Foods were classified based on the NOVA classification system [22] and categorized as unprocessed, minimally processed, processed culinary ingredients, processed foods, and ultra processed foods (UPFs).

This point should be described in more detail. Who completed this? Were there witnesses who could confirm this measure? Otherwise, the validity is questionable.

C.- Section 2.5 of statistical analysis must be explained in more detail. And it should be rewritten because the data analysis must be completely redone.

2.5. Statistical analysis  

All analyses were performed with R statistical software, version 4.3.2. Continuous variables were expressed as means and standard deviations (if normally distributed) or as medians and ranges (if non-normally distributed). Categorical variables were expressed as proportions and percentages. The patients were divided into groups, according to their consumption of packaged and processed foods. Subgroup comparisons were pe formed with the Chi-squared test for categorical variables. A p-value less than 0.05 was 160 considered statistically significant.

-How were the subgroups made? The Chi-squared coefficient must be completed with a measure of the magnitude of the effect.

D.- Regarding the presentation of the results.

-Table 1 shows that 57% of the participants do sports. In addition to controlling age in their analyses, the authors should show the results based on whether or not they do sports because it is another very important variable that can provide a lot of information. -The authors must make contingency tables and regroup the subjects at various variable levels so that the boxes have more than five subjects. They should answer the following: What is the relationship between anxiety and depression? What variables are protective, and what variables precipitate a more severe pathology? -Te results are shown in Table 2 and Table 3, given that neither age nor the fact of playing sports is controlled has no value. -The results described in section 3.2 are not understood.

Comments on the Quality of English Language

Minor editing of English language required

Round 2

Reviewer 2 Report

Comments and Suggestions for Authors

COMMENTS TO THE AUTHORS 

 Reference: Manuscript ID: nutrients-2875949

I reaffirmed everything I said in the first review. The data have not been analyzed correctly in the way that should be done when there has been little or little experimental control.

Please, the authors, reconsider all the suggestions I made.

Redo the data analysis and present it as the journal you sent the article deserves.

The data analysis section. It is unacceptable

You cannot send such huge tables. Segment the results and present them by semantic cluster. The results cannot be seen well and are not aesthetic.

Please look at Tables 2, 4, and 5. Proportions whose data boxes have a frequency of less than 5 cannot be compared.

Comments on the Quality of English Language

Minor editing of English language required

Author Response

Dear reviewer,

thank you for the review.

We are sorry to understand that we have not met your requests from the previous review.

However, the data analysis in our paper does not differ significantly from others we carried out in other studies.

Basically, we started from a descriptive analysis of the sample, followed by the comparison of patients according to the level of intake of packaged and processed food.

We are aware of the many limits of this retrospective study, including the lack of control, which calls instead for prospective controlled studies, yet we also believe that a more detailed and constructive argumentation may guide us towards a paper with an acceptable methodology.

The tables try to report the many variable analyzed in an ordered manner, for better readability.

Considering your last remark, when the expected frequencies in the contingency tables are less than 5, the Fisher exact test is used, as appropriate.